# Development and validation of the sedentary behavior regulation scale in Korean Adults Population

Mi Hwa Won[1], Sun-Hwa Shin[2]*

1 Department of Nursing, Wonkwang University, Iksan, Korea, 2 Department of Nursing, College of Nursing, Sahmyook University, Seoul, Korea

☯ These authors contributed equally to this work.
* shinsh@syu.ac.kr

## Abstract

Sedentary behavior is a well-established independent risk factor for numerous chronic diseases. However, validated tools to assess the psychological and behavioral regulation of sedentary time remain limited. This study aimed to develop and validate a self-report scale for assessing sedentary behavior regulation in Korean adults. This scale was developed using a comprehensive multiphase process. First, items were generated based on a comprehensive literature review and expert consultation, followed by a content validity assessment. In total, 600 Korean adults were recruited. Construct validity was examined using exploratory factor analysis (EFA) and confirmatory factor analysis (CFA). Criterion validity was evaluated by assessing the correlation between the new scale and the Global Physical Activity Questionnaire (GPAQ). Finally, reliability was assessed through internal consistency and test-retest reliability. The final instrument, the Sedentary Behavior Regulation Scale (SBRS), consisted of 12 items across two distinct sub-factors: sedentary behavior management and environmental support and active movement in sedentary contexts. Criterion-related validity was supported by a small but significant positive correlation with physical activity ($r = .19$, $p < .001$) and a small to moderate negative correlation with sedentary time ($r = -.33$, $p < .001$). The scale also exhibited excellent internal consistency (Cronbach's $\alpha = .87$, McDonald's omega $= .87$) and good test-retest reliability (intraclass correlation coefficient [ICC] $= .85$). Overall, the SBRS demonstrated preliminary evidence of reliability and validity for assessing behavioral and environmental strategies to reduce sedentary time. This study advances our understanding of sedentary behavior beyond simple activity measurements and provides a valuable foundation for developing targeted public health interventions to mitigate sedentary time.

**Data availability statement:** All relevant data are within the manuscript and its Supporting Information files.

**Funding:** This paper was supported by the Sahmyook University Research Fund in 2025. There was no additional external funding received for this study.

**Competing interests:** The authors declared that no conflict of interest.

# 1. Introduction

## 1.1. Background

Sedentary behavior has become pervasive in modern society, posing a significant and independent risk factor for physical and mental health [1]. This behavior is defined as any waking behavior characterized by an energy expenditure of 1.5 metabolic equivalents (METs) or less while sitting, reclining, or lying down. A growing body of evidence recognizes sedentary behavior as an independent health risk factor, distinct from the mere lack of physical activity [2]. Recent global guidelines and meta-analytic evidence have further reinforced sedentary behavior as a standalone behavioral risk factor, irrespective of engagement in moderate-to-vigorous physical activity [3,4].

Previous studies indicate that U.S. adults spend an average of 9.5 hr per day on sedentary behavior, with approximately 47% of this time occurring during leisure and work activities. A substantial proportion of leisure time (82%) is spent on screen-based activities, including watching television, using the Internet, or working on computers [5]. Similarly, in Korea, the 2021 Korea National Health and Nutrition Examination Survey showed high levels of sedentary behavior among adults, with an average of 8.9 sedentary hours per day and 63.7% sitting for at least eight hours daily. Sedentary behavior was more prevalent among adults aged 65 years and older than among those aged 19–64 years, highlighting sedentary lifestyle as a significant public health concern in Korea [6]. These trends are closely associated with societal and environmental shifts, including rapid industrialization, the proliferation of digital devices, and changes in transportation and workplace environments [7,8]. These figures should be seen not only as statistics but also as critical public health warning signals. Importantly, they indicate that sedentary behavior is structurally embedded in daily life rather than reflecting a temporary lifestyle choice, underscoring the need for sustainable behavioral regulation strategies.

Research has consistently demonstrated that increased sedentary behavior is a major risk factor for the development of various chronic diseases. Prolonged and uninterrupted sedentary time triggers a cascade of physiological changes, including insulin resistance, vascular dysfunction, altered carbohydrate metabolism, and chronic low-grade inflammation [2]. It is also associated with a shift from oxidative to glycolytic muscle fibers, reduced cardiopulmonary function, and decreased muscle and bone masses. Furthermore, it contributes to increased body and visceral fat, elevated blood lipid levels, and heightened inflammatory response [3]. Among middle-aged and older adults, sedentary behavior has been associated with cardiovascular diseases, type 2 diabetes, obesity, and certain cancers [7–9]. Emerging longitudinal and meta-analytic evidence further suggests that excessive sedentary behavior independently predicts all-cause mortality, even after controlling physical activity levels [10,11]. Beyond physical health, individuals with higher levels of sedentary behavior often experience social isolation, limitations in daily activities, and weakened physical function, which can lead to mental health problems such as depression, anxiety, and cognitive decline [3,12,13]. A systematic review found that

mortality risk increases when sedentary behavior exceeds 7.5–9 hours per day, with a particularly pronounced rise beyond 9.5 hours [11]. Ultimately, prolonged sedentary behavior poses a substantial public health threat by simultaneously undermining physical, psychological, and social well-being and reducing overall quality of life [3,12,13].

Valid assessment instruments are crucial for accurately assessing and managing the risks of sedentary behavior. However, most existing self-report surveys rely on a single question about the "total time spent sitting," a format that is highly susceptible to underreporting and inaccuracy [13]. Although objective instruments such as accelerometers provide high accuracy, they are often costly and inconvenient for large-scale use [14]. More critically, both subjective and objective measures focus primarily on quantifying sedentary time and fail to capture an individual's ability to consciously regulate, interrupt, or prevent sedentary behavior—capacities essential for effective management. Furthermore, these instruments frequently lack cross-cultural validity, a significant concern given that behaviors such as Korea's traditional floor-sitting culture are not typically accounted for in tools developed abroad [15,16]. In the Korean context, where prolonged sitting is especially concentrated among office workers and other sedentary occupational groups and is associated with elevated metabolic risk, a culturally sensitive self-report scale that assesses not only sedentary duration but also the ability to interrupt and regulate sitting may provide a practical basis for identifying high-risk groups and developing tailored behavioral interventions for everyday and workplace settings [17,18]. This conceptual and practical gap highlights the need for a theoretically grounded instrument that assesses sedentary behavior regulation rather than sedentary duration alone.

## 1.2. Aims of the current study

This study aimed to explore the conceptual meaning and behavioral characteristics of sedentary behavior regulation and develop and validate a multidimensional scale that captures individual differences in regulatory capacity. By focusing on self-regulatory processes rather than on sedentary duration alone, the proposed instrument seeks to address a critical limitation in existing measurement approaches. Ultimately, the Sedentary Behavior Regulation Scale (SBRS) is intended to provide a precise assessment tool to inform the design, tailoring, and evaluation of long-term sedentary behavior prevention and intervention programs across clinical and community settings.

## 1.3. Conceptual framework

Grounded in the principles of Self-Determination Theory (SDT), this study was designed to develop a new Sedentary Behavior Regulation Scale (SBRS). SDT is a prominent motivation theory that emphasizes the importance of internal psychological resources in personality development and behavioral regulation [19,20]. This theory posits that when basic psychological needs (competence, autonomy, and relatedness) are satisfied, intrinsic motivation is enhanced, thereby facilitating long-term behavioral change and positively influencing outcomes [14,19,21]. Consistent with this framework, sedentary behavior regulation is conceptualized as an autonomously motivated process involving self-awareness of sedentary patterns, active behavioral practices to interrupt or reduce sedentary time, and preventive environmental structuring to minimize prolonged sitting. Accordingly, SBRS is theoretically defined as an individual's conscious and active process of managing sedentary behavior in daily life. Operationally, this construct is represented by the total score obtained from the developed scale, with higher scores indicating a greater capacity for effective sedentary behavior regulation.

## 2. Methods

### 2.1. Research design

This study employed a methodological research design to develop and validate a self-report instrument for assessing sedentary behavior regulation in adults.

## 2.2. Research participants and sampling

The participants in this study were adults aged 20–59 years residing in South Korea. A proportional stratified sampling method was used to recruit a sample reflective of the national population, based on data from the Korean National Statistical Office. Stratification variables included region, sex, and age. This strategy was adopted to enhance representativeness and reduce sampling bias, consistent with recommendations for scale development studies targeting general adult populations [22]. Within each stratum, eligible participants were randomly selected from the EMBRAIN online research panel using a computerized randomization procedure to ensure proportional allocation while maintaining random selection.

Participants were recruited via EMBRAIN, a representative online research panel in South Korea consisting of a diverse pool of voluntarily registered adults. To ensure a nationally representative sample, the panel is continuously expanded and demographically balanced. To enhance data integrity, EMBRAIN implements stringent quality control measures, including identity verification, duplicate account detection, and the use of attention-check items. Furthermore, we monitored response times and excluded inconsistent or patterned responses to mitigate careless responding. Only participants who met the eligibility criteria and passed all quality control protocols were included in the final analysis.

The inclusion criteria were adults aged 20–59 years who resided in South Korea, were able to read and understand Korean, engaged in sedentary activities as part of their daily work or leisure, and voluntarily consented to participate after receiving an explanation of the study's purpose. Exclusion criteria included self-reported cognitive impairment, intellectual disability, severe psychiatric or neurological conditions that could interfere with comprehension or self-reporting, and inability to complete an online questionnaire independently. These criteria were specified to clearly define the target population for which the Sedentary Behavior Regulation Scale was intended to be valid and reliable, consistent with COSMIN recommendations regarding population validity [23].

The required sample size for the validation process was determined based on established methodological guidelines for factor analysis. For exploratory factor analysis (EFA), a minimum ratio of 10 participants per item was applied, following the recommendations of DeVellis and Thorpe [22]. This resulted in a minimum sample size of 320 participants with 32 initial items. For confirmatory factor analysis (CFA), a sample size between 200 and 400 was considered appropriate based on the structural equation modeling guidelines [24]. Accordingly, a total target sample size of 600 participants was established to allow for independent samples for EFA and CFA.

After data collection, the full sample was randomly divided into two independent subsamples using simple random allocation, with one subsample assigned to the EFA and the other to the CFA. To assess the comparability of the two subsamples, equivalence in key demographic characteristics relevant to the factor analysis, including age, sex, and region, was examined. No statistically significant differences were observed between the EFA and CFA subsamples, supporting their structural equivalence. Although more advanced procedures such as the SOLOMON method have been proposed to optimize subsample equivalence [25], the combination of proportional stratified sampling, random allocation, and empirical verification of equivalence has been deemed methodologically adequate and consistent with current best practices in scale development research.

## 2.3. Research procedure

This study followed an eight-step methodological procedure for instrument development and validation based on the framework proposed by DeVellis and Thorpe [22].

### 2.3.1. Step 1. Defining the construct.
The initial step involved defining the construct of sedentary behavior regulation through a comprehensive literature review and qualitative field study. This phase was conducted by a team comprising three researchers (two nursing researchers with expertise in scale development and one psychology researcher), all of whom participated in the construct definition process. The theoretical phase involved an analysis of 24 research articles on instrument development using databases such as the Research Information Sharing Service (RISS), Korea Institute

of Science and Technology Information System (KISIS), PubMed, and the Cumulative Index to Nursing and Allied Health Literature (CINAHL). Previous domestically and internationally developed tools measuring physical activity and sedentary behavior were reviewed [14,26–34]. Each researcher independently examined the literature and proposed preliminary conceptual domains related to the regulation of sedentary behavior. Consensus on the construct definition and its core components was reached through iterative discussions until full agreement was achieved among all researchers. Through this process, self-awareness, behavioral practices, and preventive environmental design were identified as the core components of autonomous motivation for regulating sedentary behavior.

In the field study phase, individual interviews were conducted with eight working adults (men and women) to confirm the attributes of sedentary behavior control identified in the theoretical study. Two researchers independently reviewed the interview transcripts using a directed content analysis approach guided by the preliminary theoretical framework. Meaning units related to sedentary behavior regulation were coded and grouped into categories corresponding to the identified components. After independent coding, discrepancies were resolved through discussion until consensus was reached. The finalized qualitative categories were then compared with the theoretically derived components, and areas of convergence were used to refine construct definitions. To ensure traceability, qualitative categories served as direct sources of candidate items, allowing each item to be linked to both theoretical and empirical evidence.

Findings from the literature review and interviews were integrated to refine the construct definition and guide the generation of the initial SBRS items.

**2.3.2. Step 2. Generating an item pool.** An item pool reflecting each component was developed through a comprehensive analysis of theoretical and qualitative findings. All three researchers independently generated candidate items corresponding to each identified component. These items were then compiled into a single pool and overlapping or redundant items were identified. A structured consensus procedure was applied, during which the research team reviewed each item for conceptual relevance, clarity, and consistency with the construct definition. Items were retained, revised, or removed based on unanimous agreement among the researchers. This consensus-based approach ensured that the initial item pool adequately represented the content domain of sedentary behavior regulation in accordance with the recommended practices for establishing content validity [22,23].

**2.3.3. Step 3. Determining the measurement format.** This study used a 5-point Likert scale, with the response range being 1 (strongly disagree), 2 (disagree), 3 (neutral), 4 (agree), or 5 (strongly agree).

**2.3.4. Step 4. Expert review of the initial item pool.** A panel of seven experts was convened to evaluate the content validity of the initial items. The panel comprised three head nurses and nursing education team leaders with over 10 years of clinical experience, a full-time cardiology physician, a sports medicine professor, a psychology professor, and a nursing professor (S1 Table). Each expert rated item relevance using a 4-point scale ranging from 1 (not at all valid) to 4 (very valid). The item content validity index (I-CVI) was calculated based on these ratings, and only items with an I-CVI of .80 or higher were retained. To adjust for chance agreement among experts, modified kappa statistics were calculated for each item based on the I-CVI and the number of experts, with values interpreted according to established benchmarks indicating substantial to excellent agreement beyond chance [35]. Scale-level content validity was further examined using the S-CVI/Ave. Quantitative indices and structured qualitative expert comments on relevance, clarity, and comprehensiveness were jointly considered and items were revised or removed through consensus as needed.

**2.3.5. Step 5. Evaluation of face validity.** A pilot study was conducted to quantitatively evaluate the comprehensibility of the preliminary items in terms of face validity. Ten adults aged 20–59 years were recruited to assess item clarity and ease of understanding. Participants rated their comprehension of each item on a 4-point scale ranging from 1 (difficult to understand) to 4 (very easy to understand). Items with mean comprehensibility scores below 3.0 were reviewed and revised to improve clarity. This quantitative decision rule was applied uniformly to all items, and only items that met the predefined comprehensibility criteria were retained for subsequent large-scale administrations.

**2.3.6. Step 6. Administration of the development sample.** Data was collected through an online panel survey conducted by EMBRAIN Co., Ltd. (Seoul, Republic of Korea), a company specializing in online surveys. From July 17 to July 22, 2025, invitation emails were randomly sent to 7,977 of the company's 1.78 million panelists using a sampling method based on population proportionality. Of these, 1,148 completed the survey. Among them, 157 dropped out and 118 were excluded for failing to meet the eligibility criteria, resulting in 773 complete responses. An additional 173 responses were excluded because of insincerity, such as selecting the same answer consecutively on a Likert scale or completing the survey in less than one minute. This process yielded a final sample of 600 valid responses.

**2.3.7. Step 7. Item evaluation.** Construct validity was established sequentially using correlation analysis, EFA, and CFA. Criterion-related validity was assessed by analyzing the correlations between the SBRS and the Korean version of the Global Physical Activity Questionnaire (GPAQ) [28,29]. Reliability was evaluated by assessing internal consistency using Cronbach's α and McDonald's omega, as well as test–retest reliability over a two-week interval with a subsample of 30 adults.

**2.3.8. Step 8. Optimization of scale length.** The final length of the instrument was determined through systematic evaluation of the validity and reliability of preliminary items. This process involved removing items that did not meet established psychometric criteria, thereby ensuring that the final instrument was concise and robust.

## 2.4. Criterion-related validity

To assess criterion-related validity, a survey questionnaire consisting of three sections was administered: 10 items on general characteristics, 32 items from the newly developed SBRS, and 16 items from the Korean version of the GPAQ [29]. The GPAQ, developed by the World Health Organization (WHO) in 2002, is a widely used tool for assessing physical activity and sedentary behavior [28]. The K-GPAQ has demonstrated acceptable test–retest reliability and criterion-related validity among Korean adults, with prior validation studies reporting moderate to good reliability across physical activity domains (Spearman's $r = .27–.70$; $\kappa = .30–.67$) and significant, albeit weak, correlations with accelerometer-measured physical activity and sedentary behavior ($r = .18–.34$) [29]. The GPAQ assesses the frequency (days per week) and duration (minutes per day) of vigorous physical activity, moderate-intensity physical activity, and travel over the past seven days, as well as average daily sedentary time. Physical activity levels were calculated by converting activities to MET-minutes per week in accordance with the official GPAQ analysis guide [28]. Vigorous activity was assigned 8 METs, whereas moderate activity and mobility were assigned 4 METs, with higher scores indicating a higher level of physical activity. In contrast, sedentary behavior was measured as the average daily sitting time, with a higher score indicating a higher level of sedentary behavior.

## 2.5. Ethical considerations

This study was conducted in accordance with ethical guidelines and received approval from the Institutional Review Board (IRB No: SYU 2025-03-030-001) in March 2025 prior to data collection. Research was conducted through a specialized online survey agency (EMBRAIN) that sent emails to adult panelists (aged 20–59 years) explaining the study's purpose, procedures, and participation methods. All participants provided informed consent before participating in the survey. The consent statement clearly indicated that participation was voluntary and that participants could withdraw at any time without penalty. Only those who expressed interest in participating were required to fully understand the explanation and then electronically consent to participate in the study by checking the consent box online. The system was designed to ensure that the survey was completed only after consent was obtained. This process served as a substitute for written consent, and all participants freely decided whether to participate. To ensure confidentiality, all collected data were processed anonymously, and personal information was not provided. All participants who completed the survey received a gift card as a reward.

 

## 2.6. Data analysis

Data analysis was conducted using SPSS 26.0 and AMOS 23.0 software (IBM Corp., Armonk, NY, USA). Prior to analysis, incomplete responses were removed, and only complete responses were included in EFA, CFA, and reliability analysis. Descriptive statistics were calculated for participants' general characteristics. Item analysis involved checking the means, standard deviations, skewness, and kurtosis values of the initial items. To assess the assumption of normality required for maximum likelihood estimation, univariate normality was evaluated by inspecting skewness and kurtosis values for all items, which fell within acceptable ranges. Correlation coefficients between the total score and each item were calculated, and items with correlations below .30 were examined.

Construct validity was evaluated using a two-step factor analytic approach. Prior to factor analyses, chi-square tests were conducted to examine the equivalence between the EFA (n = 320) and CFA (n = 280) groups, confirming the homogeneity of the two subsamples. EFA was conducted using Pearson correlation matrices, as the items were measured on a 5-point Likert scale and demonstrated acceptable normality based on skewness and kurtosis values. A maximum likelihood extraction method with Promax rotation was applied. Sampling adequacy was assessed using the Kaiser–Meyer–Olkin (KMO) measure and Bartlett's test of sphericity prior to factor extraction. Factor retention decisions were informed by multiple criteria, including eigenvalues greater than 1.0 and inspection of the scree plot.

CFA was subsequently performed using maximum likelihood estimation to assess the fit of the structural model. Model fit was evaluated using absolute ($\chi^2$/df, RMSEA, and SRMR) and incremental fit indices (CFI and TLI). Convergent validity was confirmed by examining correlations, construct reliability (CR), and average variance extracted (AVE) among the sub-factors. Discriminant validity was evaluated using the Fornell–Larcker criterion [36] and the heterotrait–monotrait (HTMT) ratio. HTMT values were computed based on item-level correlations, with values below the .85 threshold considered indicative of adequate discriminant validity [37].

Finally, criterion-related validity was assessed by calculating correlation coefficients between the SBRS and the Korean version of the GPAQ. Reliability was evaluated using Cronbach's α and McDonald's omega (ω) for internal consistency, as well as the intraclass correlation coefficient (ICC) for test–retest reliability.

## 3. Results

### 3.1. Item generation

A review of previous studies on sedentary behavior revealed that existing instruments are primarily limited to specific populations or rely on recording methods to assess sedentary time [28–34]. A key limitation of these instruments is their inability to capture the self-regulatory mechanisms underlying sedentary behavior. To address this gap, consultation was undertaken with a subject matter expert in psychological instrument development. Based on the comprehensive literature review, an initial pool of 55 items was generated. Following a series of rigorous content reviews, 4 items with overlapping content or low relevance to sedentary behavior regulation were removed to prevent participant confusion, resulting in a final set of 51 items.

### 3.2. Content validity

Following initial item generation, the 51 items underwent rigorous evaluation by a panel of seven experts to establish content validity (S1 Table). Items were excluded if they demonstrated a low item-level content validity index (I-CVI ≤ .80), conceptual redundancy, or insufficient relevance to the study's objectives (S2 Table). At this stage, the scale-level content validity index calculated using the averaging method (S-CVI/Ave) for the initial item pool was .87, indicating good overall content validity.

After item refinement and reduction to 32 items, the S-CVI/Ave increased to .96. A second round of content validity testing was conducted with a panel of five experts to reexamine the remaining items. All items demonstrated I-CVI values

of .80 or higher, leading to their final retention. The modified kappa values ranged from .85 to .99, indicating excellent agreement beyond chance for all items.

### 3.3. Pilot test

The pilot study evaluated 32 items in 10 adults aged 20–50 years (mean age = 45.2 years). The survey took 5–6 min to complete, and most respondents rated the items as easy to understand.

### 3.4. General characteristics of the survey participants

There were 600 survey participants, 312 men (52.03%) and 288 women (48.0%). The largest age group comprised participants in their 50s (n = 182; 30.3%). To establish construct validity, the total sample was randomly divided into two independent subsamples for EFA (n = 320) and CFA (n = 280). To verify subsample comparability, main demographic characteristics, including age and sex, were examined between the EFA and CFA groups. No statistically significant differences were found, indicating that the two subsamples were demographically equivalent and suitable for independent factor analytic procedures (Table 1).

### 3.5. Construct validity

**3.5.1. Item analysis.** Analysis was performed on the 32 preliminary items. Item means ranged from 2.02 to 4.05, and the skewness (−0.96–0.89) and kurtosis (−0.90–1.59) values met the criteria for a normal distribution. As a result of the item-to-total correlation analysis, seven items with a correlation coefficient (r) of less than .30 were removed due to their low discriminatory power, resulting in the selection of 25 items for further analysis. In particular, the self-awareness items were removed because their item–total correlation coefficients were .30 or lower, indicating insufficient psychometric performance for retention in the final factor structure.

**3.5.2. EFA.** An EFA was performed on the remaining 25 items to examine the underlying factor structure. The suitability of the data for factor analysis was confirmed using a KMO value of .93 and a statistically significant Bartlett's test of sphericity ($\chi^2$ = 3831.94, df = 300, $p$ < .001). Both eigenvalues greater than 1.0 and scree plot inspection supported a two-factor solution. During the analysis, two items (x8 and x19) were removed because their communality values were below .30. An additional four items (x17, x23, x24, and x25) were removed because their factor loadings were below .40, and three items (x13, x21, and x30) were removed because of cross-loadings, defined as factor loadings of .30 or higher on two factors in the pattern matrix. The content and psychometric properties of the excluded items, including item–total correlation coefficients, communalities, and factor loadings, are presented in S3 Table.

Ultimately, 16 items were selected and loaded onto two distinct factors. The final model's KMO value was .91, and Bartlett's test of sphericity remained statistically significant ($\chi^2$ = 2368.24, df = 120, $p$ < .001). All retained items exhibited factor loadings of .44 or higher, with communality values of .32 or greater. The two factors jointly accounted for 54.31% of the total variance (Table 2). The overall reliability of the final 16-item instrument was excellent (Cronbach's α = .90). The two sub-factors were named based on the content of their loaded items: Factor 1, "sedentary behavior management and environmental support"; and Factor 2, "active movement in sedentary contexts."

**3.5.3. CFA.** CFA was conducted to examine the structural relationships between the two factors derived from EFA. Model fit was evaluated using established cut-off criteria: $\chi^2$/df values below 3.0, CFI and TLI values ≥ .90, SRMR ≤ .08, and RMSEA ≤ .08 were considered indicative of acceptable model fit, while lower $\chi^2$/df and RMSEA values and higher CFI and TLI values indicated good fit. The initial CFA model demonstrated suboptimal fit, with a $\chi^2$/df value of 3.43, and other indices that did not meet recommended standards (CFI = .88, TLI = .86, SRMR = .084, RMSEA = .093, 90% CI [.083, .104]).

To avoid purely data-driven modifications, an iterative model refinement process was conducted based on modification indices (MI) and theoretical considerations, including conceptual overlap and redundancy among items with highly similar

**Table 1. General Characteristics of the Participants (N = 600).**

| Characteristics | Categories | Total (n = 600) n(%) | EFA (n = 320) n(%) | CFA (n = 280) n(%) | χ² (p) |
|---|---|---|---|---|---|
| Sex | Men | 312(52.0) | 160(50.0) | 152(54.3) | 1.10 (.167) |
| | Women | 288(48.0) | 160(50.0) | 128(45.7) | |
| Age (years) | 20~29 | 123(20.5) | 70(21.9) | 53(18.9) | 0.63 (.653) |
| | 30~39 | 133(22.2) | 74(23.1) | 59(21.1) | |
| | 40~49 | 162(27.0) | 84(26.3) | 78(27.9) | |
| | 50~59 | 182(30.3) | 92(28.8) | 90(32.1) | |
| Residential area | Seoul | 120(20.0) | 60(18.8) | 60(21.4) | 1.40 (.705) |
| | Gyeonggi-do | 158(26.3) | 82(25.6) | 76(27.1) | |
| | Metropolitan City | 157(26.2) | 89(27.8) | 68(24.3) | |
| | Medium-sized Cities | 165(27.5) | 89(27.8) | 76(27.1) | |
| Academic background | High school graduate | 121(20.2) | 63(19.7) | 58(20.7) | 0.45 (.798) |
| | University graduate | 421(70.2) | 228(71.3) | 193(68.9) | |
| | Graduate school graduate | 58(9.7) | 29(9.1) | 29(10.4) | |
| Monthly income (KRW) | < 2,990,000 | 275(45.8) | 156(48.8) | 119(42.5) | 4.26 (.119) |
| | 3,000,000~4,990,000 | 186(31.0) | 100(31.3) | 86(30.7) | |
| | ≥ 5,000,000 | 139(23.2) | 64(20) | 75(26.8) | |
| Marital status | Unmarried | 341(56.8) | 185(57.8) | 156(55.7) | 0.27 (.621) |
| | Married | 259(43.2) | 135(42.2) | 124(44.3) | |
| Employment status | Unemployed | 125(20.8) | 63(19.7) | 62(22.1) | 0.55 (.482) |
| | Employed | 475(79.2) | 257(80.3) | 218(77.9) | |
| Hospitalization (within one year) | No | 551(91.8) | 293(91.6) | 258(92.1) | 0.07 (.882) |
| | Yes | 49(8.2) | 27(8.4) | 22(7.9) | |
| Health status | Not healthy | 155(25.8) | 78(24.4) | 77(27.5) | 1.68 (.430) |
| | Average | 289(48.2) | 162(50.6) | 127(45.4) | |
| | Healthy | 156(26.0) | 80(25) | 76(27.1) | |
| Body mass index (kg/m²) | Underweight (<18.5) | 39(6.5) | 19(5.9) | 20(7.1) | 1.32 (.726) |
| | Normal (18.5~22.9) | 243(40.5) | 134(41.9) | 109(38.9) | |
| | Overweight (23.0~24.9) | 127(21.2) | 70(21.9) | 57(20.4) | |
| | Obese (≥25.0) | 191(31.8) | 97(30.3) | 94(33.6) | |
| Alcohol consumption | Never drink | 355(59.2) | 190(59.4) | 165(58.9) | 0.48 (.788) |
| | 1-2 times a week | 163(27.2) | 89(27.8) | 74(26.4) | |
| | 3 or more times a week | 82(13.7) | 41(12.8) | 41(14.7) | |
| Smoking | Never smoke | 467(77.8) | 250(78.1) | 217(77.5) | 0.79 (.675) |
| | Less than 10 cigarettes per day | 58(9.7) | 33(10.3) | 25(8.9) | |
| | 11 or more cigarettes per day | 75(12.5) | 37(11.6) | 38(13.6) | |

EFA = Exploratory factor analysis; CFA = Confirmatory factor analysis.

behavioral intent. This process led to the deletion of four items (x28, x31, x15, and x18), resulting in a final model with 12 items (Fig 1). The final revised model demonstrated good fit to the data, with all indices meeting recommended standards: $\chi^2/df = 2.51$, CFI = .93, TLI = .92, SRMR = .056, and RMSEA = .074 (90% CI [.058, .089]).

All standardized factor loadings (β) were statistically significant, and their 95% bootstrap confidence intervals did not include zero (Table 3). Reliability was confirmed, as both factors showed CR values above .70. However, the AVE was below the recommended .50 benchmark for both factors, indicating insufficient convergent validity. This suggests that

**Table 2. Result of Exploratory Factor Analysis (N = 320).**

| Item | Factor loading | Communality | Eigen -values | VE (%) | CV (%) | Cronbach's α |
|---|---|---|---|---|---|---|
| Factor 1 | | | 6.54 | 40.85 | 40.85 | .88 |
| x31 | 0.95 | 0.74 | | | | |
| x32 | 0.92 | 0.75 | | | | |
| x28 | 0.82 | 0.58 | | | | |
| x29 | 0.59 | 0.51 | | | | |
| x27 | 0.54 | 0.39 | | | | |
| x16 | 0.54 | 0.38 | | | | |
| x26 | 0.54 | 0.35 | | | | |
| x22 | 0.50 | 0.43 | | | | |
| Factor 2 | | | 2.16 | 13.47 | 54.31 | .85 |
| x10 | 0.79 | .0.57 | | | | |
| x9 | 0.78 | 0.50 | | | | |
| x12 | 0.76 | 0.63 | | | | |
| x20 | 0.61 | 0.32 | | | | |
| x11 | 0.58 | 0.36 | | | | |
| x18 | 0.49 | 0.42 | | | | |
| x14 | 0.46 | 0.35 | | | | |
| x15 | 0.44 | 0.42 | | | | |
| Total | | | | | | .90 |

VE = Variance explained; CV = Cumulative variance; Factor 1 = sedentary behavior management and environmental support; Factor 2 = active movement in sedentary contexts.

further instrumental refinement is required to enhance this aspect. The Fornell–Larcker criterion for discriminant validity was not satisfied, as the square root of AVE for Factor 1 (.67) was lower than the inter-factor correlation (r = .68, p < .001), and the value for Factor 2 (.68) was only equal to the correlation. To further evaluate discriminant validity, the HTMT value was calculated. The HTMT value between Factors 1 and 2 was .69, well below the conservative threshold of .85. These results indicate that although the two factors are moderately correlated, they represent empirically distinct constructs.

### 3.6. Criterion validity

Criterion validity was verified using the GPAQ. The total SBRS score showed a significant positive correlation with physical activity (r = .19, p < .001) and a significant negative correlation with sedentary time (r = −.33, p < .001) (Table 4). For individual sub-factors, physical activity was positively correlated with Factors 1 (r = .14, p = .011) and 2 (r = .18, p = .001). Conversely, sedentary time showed a significant negative correlation with factors 1 (r = −.28, p < .001) and 2 (r = −.30, p < .001), further supporting the validity of the instrument.

### 3.7. Reliability

Cronbach's α for the entire instrument was .87, indicating high internal consistency. Factor 1 had a value of .83, and Factor 2 had a value of .82, both of which exceeded the recommended benchmark of .80. The McDonald's omega (ω) coefficient for the entire instrument was .87, while Factor 1 and Factor 2 showed ω values of .81 and .82, respectively, indicating satisfactory internal consistency under a congeneric measurement model. Furthermore, the ICC for test-retest reliability was .85 for the total instrument, with Factor 1 at .81 and Factor 2 at .82. These high values confirmed the stability of the instrument over time.

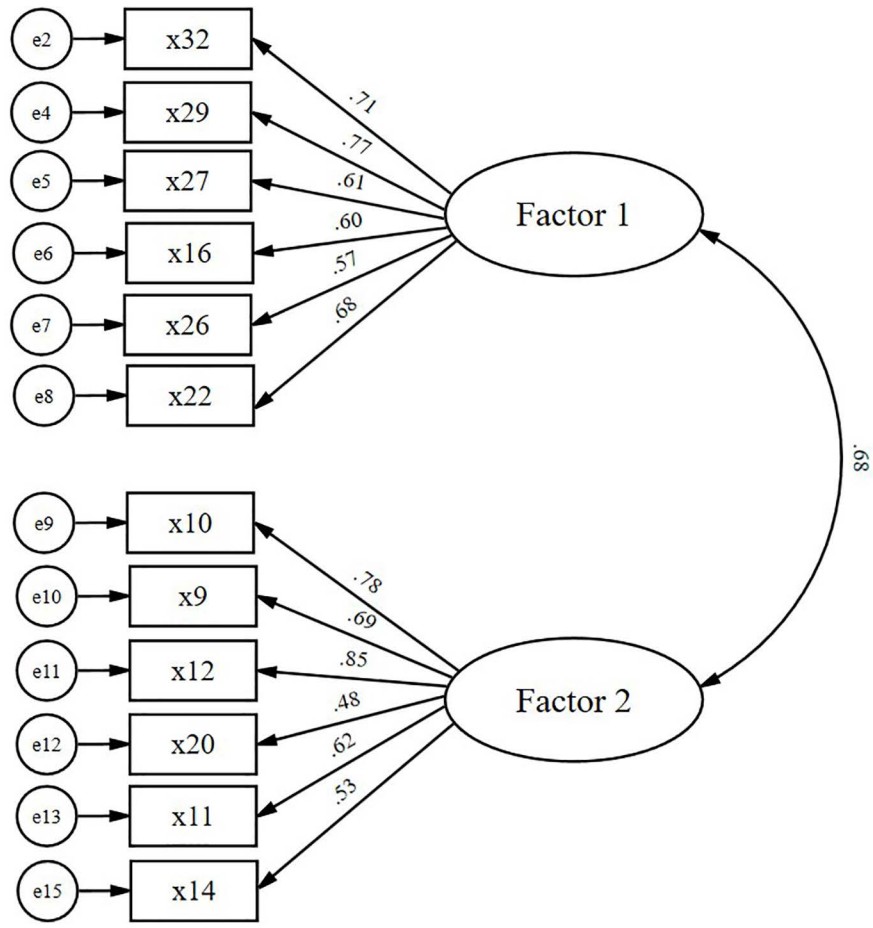

**Fig 1. Confirmatory factor analysis of the measurement model.**

### 3.8. Optimization of the scale

This study finalized the SBRS, a 12-item instrument that assesses an individual's ability to manage sedentary behaviors. The items were structured on a 5-point Likert scale ranging from 1 (strongly disagree) to 5 (strongly agree), with higher scores indicating more effective sedentary behavior management. The total SBRS score was calculated by summing all 12 items, yielding a possible range of 12–60. The final item composition by sub-factor was as follows: Factor 1 (sedentary behavior management and environmental support) comprised items 1–6, and Factor 2 (active movement in sedentary contexts) comprised items 7–12 (S4 Table). At the current stage of scale development, no empirical cut-off scores have been proposed; thus, SBRS scores are intended to be interpreted relative to sample distributions or for comparative purposes across individuals, groups, or time points.

## 4. Discussion

This study aimed to develop a scale to measure sedentary behavior regulation in adults and to examine its psychometric validity and reliability. A 12-item scale comprising 2 sub-factors was finalized. The following discussion interprets the findings with particular attention paid to the theoretical framework and psychometric results.

**Table 3. Result of Confirmatory Factor Analysis (N = 280).**

| Factor | Item | β | B | SE | C.R. | p | 95% CI | AVE | CR |
|---|---|---|---|---|---|---|---|---|---|
| Factor 1 | x32 | 0.71 | 1 | | | | (reference) | 0.45 | 0.82 |
| | x29 | 0.77 | 1.02 | 0.09 | 11.19 | <.001 | [0.84, 1.20] | | |
| | x27 | 0.62 | 0.95 | 0.1 | 9.2 | <.001 | [0.75, 1.15] | | |
| | x16 | 0.6 | 0.82 | 0.09 | 9.06 | <.001 | [0.64, 1.00] | | |
| | x26 | 0.57 | 0.85 | 0.1 | 8.51 | <.001 | [0.65, 1.05] | | |
| | x22 | 0.68 | 0.95 | 0.09 | 10.09 | <.001 | [0.77, 1.13] | | |
| Factor 2 | x10 | 0.78 | 1 | | | | (reference) | 0.46 | 0.83 |
| | x9 | 0.69 | 0.87 | 0.08 | 11.45 | <.001 | [0.71, 1.03] | | |
| | x12 | 0.85 | 1.11 | 0.08 | 14.2 | <.001 | [0.95, 1.27] | | |
| | x20 | 0.48 | 0.55 | 0.07 | 7.8 | <.001 | [0.41, 0.69] | | |
| | x11 | 0.62 | 0.8 | 0.08 | 10.2 | <.001 | [0.64, 0.96] | | |
| | x14 | 0.53 | 0.72 | 0.08 | 8.54 | <.001 | [0.56, 0.88] | | |

Note. β = Standardized coefficient; B = Unstandardized coefficient; SE = Standard error; C.R. = Critical ratio; CI = Confidence intervals; AVE = Average variance extracted; CR = Construct reliability; Factor 1 = sedentary behavior management and environmental support; Factor 2 = active movement in sedentary contexts. All confidence intervals exclude zero.

**Table 4. Correlation between SBRS and GPAQ (N = 320).**

| Variables | SBRS | | GPAQ | | Min | Max | M±SD |
|---|---|---|---|---|---|---|---|
| | Factor 1 | Factor 2 | PA | ST | | | |
| SBRS | .89 (p<.001) | .85 (p<.001) | .19 (p<.001) | −.33 (p<.001) | 13 | 55 | 36.18±7.67 |
| Factor 1 | | .51 (p<.001) | .14 (p=.011) | −.28 (p<.001) | 6 | 28 | 15.70±4.72 |
| Factor 2 | | | .18 (p=.001) | −.30 (p<.001) | 7 | 30 | 20.48±4.09 |
| GPAQ | | | | | | | |
| PA(METs/min) | | | | −.36 (p<.001) | 0 | 46350 | 3901.88±5923.14 |
| ST (hours) | | | | | 0.13 | 20 | 8.61±4.82 |

SBRS = Sedentary behavior regulation scale; Factor 1 = sedentary behavior management and environmental support; Factor 2 = active movement in sedentary contexts; GPAQ = Global physical activity questionnaire; PA = physical activity; ST = sedentary time.

The final SBRS consists of two sub-factors: (1) sedentary behavior management and environmental support, and (2) active movement in sedentary contexts. This structure extends beyond the traditional view that conceptualizes sedentary behavior as the absence of physical activity. Previous studies have predominantly relied on instruments such as the IPAQ, GPAQ, and SBQ to assess sedentary behavior in terms of frequency and duration [28–34]. However, when sedentary behavior is evaluated solely based on cumulative sitting time, important qualitative differences—such as intentional interruptions or regulatory patterns—may be overlooked. Accordingly, the present findings support the perspective that sedentary behavior should be understood and measured as a regulatory process rather than a passive accumulation of sitting time [38]. Unlike existing instruments, the SBRS captures both individual self-regulatory behaviors and environmental conditions that facilitate or constrain sedentary behavior management. This approach allows sedentary behavior to be conceptualized as a controllable and modifiable behavior, providing a theoretical basis for intervention strategies that target regulatory mechanisms rather than sedentary time alone.

From the perspective of SDT, the two retained factors align closely with the basic psychological needs of autonomy and competence [19,20]. The first factor, sedentary behavior management and environmental support, reflects intentional

strategies through which individuals modify their environments and manage daily routines to reduce sedentary time. Behaviors such as setting goals to limit sitting, deliberately creating opportunities to stand while working, or preparing environmental cues (e.g., stretching tools) reflect autonomy through self-endorsed goal setting and decision-making. Simultaneously, these strategies reinforce competence by enabling individuals to successfully implement and sustain regulatory behaviors. Environmental adjustments that reduce discomfort and support prolonged engagement in regulatory actions further contribute to competence by facilitating effective behavioral execution. The second factor, active movement in sedentary contexts, captures immediate self-initiated actions taken in response to bodily discomfort or perceived health risks while sitting. Behaviors such as standing up during prolonged work, moving during long drives, stretching to relieve discomfort, or adjusting posture reflect competence, as they demonstrate the ability to perceive bodily cues and respond effectively. These actions also support autonomy as they are initiated voluntarily and enacted without external prompts. Collectively, this factor represents the enacted self-regulation embedded within sedentary contexts, emphasizing continuous physical activation rather than a simple reduction in sitting duration. In contrast, relatedness was not clearly captured in the final scale. This may be because sedentary behavior regulation is often enacted as an individual, self-directed process rather than through interpersonal interaction. Therefore, although the SBRS reflects the autonomy and competence dimensions of SDT, future refinement may be needed to examine whether relatedness-based aspects of regulation should also be incorporated.

This study conceptualized three core components—self-awareness, behavioral practice, and preventive environmental support—based on SDT to develop a sitting behavior regulation scale. An important theoretical implication of this study is the exclusion of self-awareness items during the factor analytic process. This interpretation was refined on the basis of the empirical findings of the present study, rather than being assumed as part of the initial conceptual framework. Although self-awareness was initially conceptualized as a core component of sedentary behavior regulation, all related items were removed because of insufficient psychometric performance. This finding suggests that self-awareness may function as a cognitive antecedent or prerequisite for regulation rather than as an independent regulatory dimension. Given the habitual and often unconscious nature of sedentary behavior, awareness alone may be insufficient to produce meaningful behavioral changes unless translated into concrete actions and environmental modifications. Accordingly, the final two-factor structure indicates that behavioral practice and environmental support constitute the most salient and measurable dimensions of sedentary behavior regulation. This interpretation is consistent with SDT, which emphasizes enacted, self-determined behavior as the primary mechanism through which motivation is translated into sustained behavioral change. From an intervention perspective, these findings imply that programs aimed at reducing sedentary behavior may be more effective if they prioritize actionable strategies and supportive environments over raising awareness.

Although the final 12 items of the SBRS do not explicitly reference floor sitting, they may still be relevant to sedentary contexts characteristic of Korean daily life. In Korea, sedentary behavior often occurs not only in chair-based settings but also in floor-seated postures during activities such as dining, leisure, and social interaction. This contextual relevance may be particularly important in Korea, where objectively measured sitting time appears to be shaped by demographic and everyday contextual correlates, suggesting that sedentary behavior assessment may benefit from considering how sitting is embedded in daily routines rather than relying solely on generic duration-based indicators [17]. For example, items addressing frequent postural adjustment, stretching, standing up in response to discomfort, and intentional movement during prolonged sitting reflect self-regulatory strategies commonly used to mitigate the physical strain associated with floor-based and low-seated positions. From this perspective, the potential cultural relevance of the SBRS may lie in its focus on how individuals manage bodily discomfort and initiate movement within seated contexts, rather than in any direct assessment of floor-sitting behavior itself. Accordingly, the scale may have relevance within the Korean sociocultural context while retaining possible applicability to other cultures with similar sedentary practices. However, whether these items function similarly across chair-based and floor-based sedentary settings remains an empirical question that should be examined in future research.

The SBRS demonstrated acceptable psychometric properties, including high internal consistency (Cronbach's alpha = .87, McDonald's omega = .87) and satisfactory test–retest reliability (ICC = .85). In addition, although the Fornell–Larcker criterion was not satisfied and the AVE values remained below the recommended threshold, the HTMT result suggested that the two factors may still be empirically distinguishable. These findings indicate that, while the constructions measured by the SBRS are empirically distinguishable, the scale captures regulatory tendencies with some degree of measurement error, necessitating a cautious interpretation of effect sizes and associations. Accordingly, the present study should be regarded as a developmental validation study providing initial evidence for the scale's reliability and validity, rather than a definitive validation.

The SBRS demonstrated a statistically significant positive correlation with physical activity and a negative correlation with sedentary time. Although the observed effect sizes were small to moderate, similar magnitudes have been reported in validation studies of self-regulation and self-efficacy measures related to sedentary behavior and physical activity, supporting the theoretical meaningfulness of the present findings [39,40]. These results suggest that the SBRS is useful for predicting actual changes in sedentary behavior and aligns with existing studies linking physical activity, sedentary time, self-efficacy, and motivational control [41]. This scale addresses a key limitation of existing physical activity instruments by directly measuring the regulatory mechanisms of sedentary behavior, thereby providing a new tool for identifying its underlying causes. As a self-report instrument, it facilitates easy assessment of lifestyle patterns and offers a practical method applicable in large-scale studies and clinical settings [40]. This practical utility is especially relevant in Korea, where prolonged sitting and sedentary occupational patterns have been associated with metabolic syndrome, indicating that a self-report instrument capable of identifying deficits in sedentary behavior regulation may support early screening and tailored intervention in community and workplace settings [18]. Nevertheless, future research should refine item content, collect additional data, and revalidate the scale to enhance convergent validity and strengthen its applicability in both research and practice.

## 4.1. Limitations

This study has several limitations. First, data collection relied on an online panel, which may have introduced a sampling bias toward adults familiar with digital environments, thereby limiting the generalizability of the findings to a broader population. Second, the use of self-report questionnaires may have introduced recall or social desirability biases. Future studies should incorporate objective instruments, such as accelerometers or wearable sensors, to provide a more precise assessment of sedentary behavior regulation and cross-validate self-report data. Third, the sample was restricted to Korean adults aged 20–59 years, which limits the generalizability of the findings to older adults, particularly those aged 60 years and older, who may exhibit distinct sedentary behavior patterns, health conditions, and regulatory processes. In addition, cultural specificity should be considered, as sedentary behavior and its regulation may vary across sociocultural contexts. Accordingly, future studies should validate the SBRS across diverse age groups and populations. Fourth, the AVE values for both factors did not reach the recommended threshold of .50, indicating that a substantial proportion of the variance in the observed items may be attributable to measurement error rather than latent constructs. Although the HTMT result suggested that the two factors may be empirically distinguishable, insufficient AVE may limit construct precision and reduce statistical power in subsequent analyses. In particular, associations between SBRS scores and related outcomes may be underestimated, and caution is warranted when interpreting small effect sizes or non-significant findings. Moreover, the failure to satisfy the Fornell–Larcker criterion, together with AVE values below .50, raises the possibility that the two factors may not yet be sufficiently distinct at the construct level, and this issue should be prioritized in future refinement of the scale. Taken together, these limitations indicate that the present study represents an initial, developmental validation aimed at establishing the initial psychometric properties of the SBRS rather than a definitive validation. These findings highlight the need for further refinement of item content and factor structure to enhance variance extraction and measurement precision. Furthermore, items removed because of cross-loadings should be

re-evaluated in future studies with closer attention to their theoretical relevance and substantive content, rather than being excluded solely on statistical grounds.

## 4.2. Implications and recommendations

Future research should address these limitations. First, the generalizability of the SBRS should be confirmed using larger and more diverse samples, including participants from diverse regions and occupational groups. Second, longitudinal research tracking the validity of the SBRS over time is required to clarify the causal relationship between sedentary behavior regulation levels and health indicators (e.g., obesity, metabolic syndrome, and mental health). Third, additional data should be collected to refine and revalidate the current item set, with particular emphasis on revising or replacing items with lower factor loadings to improve convergent validity and AVE. Such refinement is expected to enhance the sensitivity and statistical power of the scale in future research applications.

From an applied perspective, the current version of the SBRS may be most appropriately used as a screening or exploratory tool to capture general tendencies in sedentary behavior regulation rather than as a high-stakes diagnostic instrument. Therefore, future intervention studies should interpret SBRS scores as indicative of the relative regulatory capacity and use them to inform the development of tailored programs for obesity, metabolic syndrome, and mental health. Finally, experimental and population-specific studies—focusing on groups such as office workers, remote workers, and older adults—are needed to further validate the scale and examine the effectiveness of interventions designed to modify sedentary behavior regulation in diverse contexts.

## 5. Conclusions

This study developed and preliminarily validated the SBRS to measure sedentary behavior regulation in adults. The final instrument comprises 12 items organized into 2 sub-factors: (1) sedentary behavior management and environmental support, and (2) active movement in sedentary contexts. The SBRS uses a 5-point Likert response format, with higher scores indicating a more effective ability to regulate sedentary behavior. The SBRS demonstrated acceptable psychometric properties, including good internal consistency and stability. Criterion validity was supported by statistically significant associations with physical activity and sedentary time, with effect sizes ranging from small to moderate. By conceptualizing sedentary behavior as modifiable and providing preliminary evidence for the SBRS as a measurement tool, this study offers academic and practical value. The SBRS serves as a foundational instrument for designing and evaluating future intervention programs aimed at reducing sedentary behavior in adults. Ultimately, it can contribute crucial data for establishing public health policies and advancing health promotion strategies.

## Supporting information

**S1 Table. Characteristics of the Expert Panel.**
(DOCX)

**S2 Table. Content Validity Indices and Item Reduction Process.**
(DOCX)

**S3 Table. Items Excluded Based on Item-Total Correlation Coefficient and EFA Results.**
(DOCX)

**S4 Table. Final Version of the Sedentary Behavior Regulation Scale (SBRS).**
(DOCX)

## Acknowledgments

None.

## Author contributions

**Conceptualization:** Mi Hwa Won, Sun-Hwa Shin.

**Formal analysis:** Mi Hwa Won, Sun-Hwa Shin.

**Funding acquisition:** Sun-Hwa Shin.

**Investigation:** Sun-Hwa Shin.

**Methodology:** Mi Hwa Won, Sun-Hwa Shin.

**Project administration:** Sun-Hwa Shin.

**Supervision:** Mi Hwa Won, Sun-Hwa Shin.

**Validation:** Mi Hwa Won, Sun-Hwa Shin.

**Visualization:** Sun-Hwa Shin.

**Writing – original draft:** Mi Hwa Won, Sun-Hwa Shin.

**Writing – review & editing:** Mi Hwa Won, Sun-Hwa Shin.

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
