## [Decision Letter · Decision Letter 0]

19 Jan 2026

Dear Dr. Shin,

Thank you for submitting your manuscript to PLOS ONE. After careful consideration, we feel that it has merit but does not fully meet PLOS ONE’s publication criteria as it currently stands. Therefore, we invite you to submit a revised version of the manuscript that addresses the points raised during the review process.

We look forward to receiving your revised manuscript.

Kind regards,

Muhammad Shawqi, MD, MSc

Guest Editor

PLOS One

Journal Requirements:

[This paper was supported by the Sahmyook University Research Fund in 2025.].

3. Thank you for stating the following in your manuscript:

[This paper was supported by the Sahmyook University Research Fund in 2025.]

[This paper was supported by the Sahmyook University Research Fund in 2025.]

5. Please upload a copy of Supporting Information File which you refer to in your text on page 26.

Reviewers' comments:

Reviewer's Responses to Questions

**Comments to the Author**

1. Is the manuscript technically sound, and do the data support the conclusions?

Reviewer #1: Yes

Reviewer #2: Partly

2. Has the statistical analysis been performed appropriately and rigorously?

Reviewer #1: Yes

Reviewer #2: No

3. Have the authors made all data underlying the findings in their manuscript fully available?

Reviewer #1: Yes

Reviewer #2: No

4. Is the manuscript presented in an intelligible fashion and written in standard English?

Reviewer #1: Yes

Reviewer #2: No

Reviewer #1: Dear authors,

Thank you for allowing me to review your manuscript. The development of new measurement instruments is an area of particular interest and scientific value; overall, your work is relevant and presents a reasonable methodological basis. However, I consider that the manuscript can be improved, especially in the Methods section and in the transparency of the content validity process. Below I provide observations intended to strengthen the quality of the study and its reporting.

1. Introduction

The introduction contextualises the construct appropriately. However, it would be advisable to update some references and further strengthen the justification for the instrument in terms of the knowledge gap and practical need.

2. Methods

You indicate that you followed the framework proposed by DeVellis and Thorpe, which is appropriate. Nevertheless, to meet current standards (COSMIN and good practices in scale development), several aspects should be expanded and clarified:

2.1. Participants and sampling

* It is necessary to specify the inclusion and exclusion criteria more precisely, as they define the population for which the instrument can be considered valid and reliable.

* Please describe in greater detail the sampling/selection procedure (how strata were defined, how randomisation was performed, and how participants were assigned to the EFA and CFA). You state that you split the sample to conduct EFA and CFA, which is appropriate. However, it would be advisable to describe the splitting procedure explicitly (e.g., simple random allocation) and, if possible, consider methods that ensure equivalent subsamples on characteristics relevant to factor analysis, such as the SOLOMON method (Lorenzo-Seva, 2022), or at least report evidence of equivalence between subsamples. Lorenzo-Seva, U. (2022). SOLOMON: A method for splitting a sample into equivalent subsamples in factor analysis. Behavior Research Methods, 54(6), 2665–2677 . https://doi.org/10.3758/s13428-021-01750-y. https://doi.org/10.3758/s13428-021-01750-y

* Please clarify what the EMBRAIN panel is (characteristics, coverage, quality control mechanisms, representativeness, and potential biases), as this affects external validity.

2.2. Construct definition and item generation

* Please indicate how many researchers participated in defining the construct and generating the initial item pool, and describe the consensus procedure.

* If initial interviews or qualitative data collection were conducted, please describe more clearly how that information contributed to instrument development (how it was coded, how it was transformed into items, and how traceability was ensured).

2.3. Content validity (critical point according to COSMIN)

Content validity is the key measurement property for new instruments. Reporting could be strengthened as follows:

* In addition to the quantitative calculation of the I-CVI, it would be advisable to explicitly incorporate a qualitative evaluation by experts and/or the target population (structured comments on relevance, clarity, and comprehensiveness) and explain how disagreements were resolved.

* Please consider reporting scale-level indices (e.g., S-CVI/Ave and, if applicable, S-CVI/UA), not only item-level indices.

* I suggest adding a correction for chance agreement to the I-CVI using the modified kappa (k*) (or, at minimum, justifying why it was not applied). This would strengthen the content validity argument.

* I attach a helpful reference:

Almanasreh E, Moles R, Chen TF. *Evaluation of methods used for estimating content validity.* Res Social Adm Pharm. 2019 Feb;15(2):214–221. doi: 10.1016/j.sapharm.2018.03.066. Epub 2018 Mar 27. PMID: 29606610.

* It would be useful to include, as supplementary material, a table describing the experts’ characteristics (profile, experience, selection criteria) and item-by-item results (I-CVI by round, decisions: retain/modify/delete).

2.4. Face validity / comprehensibility

The pilot testing is appropriate. However, to align with COSMIN, it would be desirable to describe it as an evaluation of comprehensibility and, if possible, complement it with cognitive interviews (even in a small sample). If it only remains purely quantitative, please describe decision criteria (cut-offs) and the item refinement procedure.

2.5. Psychometric analysis (EFA/CFA) and reliability

* Please clarify whether the factor analysis was based on Pearson or polychoric correlations, and justify the decision given the Likert format.

* It should be stated in the Methods that KMO and Bartlett’s test were used to assess sampling adequacy. Please specify the extraction method, the factor retention criteria (I would recommend including parallel analysis and/or a scree plot, beyond the eigenvalue > 1 criterion), and the criteria used for cross-loadings.

* For reliability, Cronbach’s α is commonly used but insufficient as the sole indicator; I recommend adding ω (omega) and justifying its use based on the final structure. Please review page 49 of: DeVellis RF, Thorpe CT. *Scale development: Theory and applications* (5th ed.). Sage, where the limitations of Cronbach’s alpha are discussed.

* When reporting CFA fit indices, please indicate the cut-off values used to define an acceptable/good model and avoid purely data-driven modifications unless they are theoretically justified.

* In addition, confidence intervals were not calculated for all estimates, which reduces the quality of the validation evidence. Please consider this aspect in future studies.

2.6. Criterion validity

If an external instrument is used as a criterion, please provide more information on its scoring system and its psychometric properties in the linguistic/cultural context of application, so that readers can interpret the magnitude and relevance of the correlations. Therefore, please provide validity and reliability data for the Korean version of the GPAQ.

3. Results

Some additional or expanded results could be presented as supplementary material (e.g., I-CVI per item and per round; item refinement decisions; response distributions in the pilot test). There are no data on item scores in the pilot test, and these could be provided as supplementary material.I also await the editor's opinion on this matter.

4. Presentation and interpretation of the final instrument

Please clarify precisely the scoring method (total score vs. subscales; minimum–maximum ranges; interpretation of higher scores) and whether any proposed cut-offs exist or whether interpretation is only relative.

I believe these comments are sufficient for a first round. Overall, I consider your results promising, but the manuscript requires substantial improvement in methodological reporting and, in particular, in the rationale and documentation of content validity. I hope these observations are helpful in strengthening your work and are received as an opportunity for improvement.

Sincerely,

Reviewer #2: This study is original in that it conceptualizes sedentary behavior not merely as a lack of physical activity but as a modifiable and regulatable behavior, which has clear public health relevance. However, several important methodological issues and areas of insufficient reporting remain. For these reasons, I believe that major revision is required before the manuscript can be considered for publication.

1. Insufficient Convergent Validity (AVE)

The average variance extracted (AVE) values for both factors are below the recommended threshold of 0.50, indicating insufficient convergent validity. Although the authors acknowledge this limitation, it remains a serious concern regarding the validity of the scale. A low AVE means that the variance of the observed variable is due to more Error than the latent variable. Therefore, there is a risk that using this measure in future studies will not provide sufficient statistical power. If the AVE is low, the discriminant validity between factors (Fornell-Larcker criteria or HTMT ratio) may also not be met.

The authors are encouraged to reconsider the current items and, if possible, collect additional data to refine the scale. Alternatively, the study could be explicitly positioned as a preliminary or developmental study, with a stronger emphasis on the need for further refinement and validation. In the Discussion section, the authors should elaborate more clearly on how insufficient AVE may affect the interpretation and application of the SBRS in research and practice.

2. Insufficient Theoretical Justification for the Factor Structure

In the conceptual framework, three core components—self-awareness, behavioral practice, and preventive environmental design—are identified as key elements of sedentary behavior regulation. However, the final validated scale consists of a two-factor structure, and all items related to self-awareness were excluded during the factor analytic process.

Although this discrepancy is briefly mentioned in the Discussion, the theoretical justification is insufficient. A deeper discussion is needed to explain why self-awareness, which was theoretically emphasized, failed to emerge as a statistically meaningful factor, and what this implies for the conceptualization of sedentary behavior regulation.

In addition, while the SBRS is described as being grounded in Self-Determination Theory (SDT), it remains unclear how the two final factors—“sedentary behavior management and environmental support” and “active movement in sedentary contexts”—correspond to the three basic psychological needs proposed by SDT (autonomy, competence, and relatedness). A clearer and more systematic explanation of this theoretical alignment is necessary.

Scale development should be a consistent process: theory (SDT), item generation (32 questions), statistical examination (EFA/CFA), and final scale (12 questions). However, the current situation seems to separate theory from the final statistical results. In particular, please consider the theoretical implications of the omission of self-awareness.

3. Insufficient Explanation of the Item Reduction Process

During the reduction from 32 initial items to the final 12-item scale, a total of 20 items were removed (9 during EFA and an additional 4 during CFA). However, the manuscript does not provide sufficient detail regarding the content of the deleted items or the specific reasons for their removal.

The authors should provide a list of the excluded items and clearly explain the rationale for deletion. In particular, items described as having “redundant content” require more concrete and transparent justification.

4. Issues Related to Sample Size and Sample Splitting

Although the rationale for the initial sample size calculation is described, the subsequent reduction from 32 to 12 items raises concerns about whether the original justification remains appropriate. This issue should be addressed explicitly.

Furthermore, regarding the division of the sample into EFA (n = 320) and CFA (n = 280) groups, it is unclear whether the split was fully random and whether key demographic characteristics (e.g., age, sex) were equivalent between the two groups. This information should be clearly reported.

In addition, the restriction of the sample to Korean adults aged 20–59 years should be more explicitly discussed as a limitation affecting the generalizability of the findings.

5. Interpretation of Effect Sizes for Criterion Validity

The correlation between the SBRS and physical activity (r = .19) is statistically significant but represents a small effect size, while the correlation with sedentary time (r = −.33) is moderate. On this basis, concluding that the scale demonstrates “strong criterion validity” appears to be an overstatement.

The authors should provide a clearer theoretical rationale explaining why correlations of this magnitude are considered appropriate or meaningful for a measure of sedentary behavior regulation, and the interpretation should be more cautious.

6. Insufficient Description of Statistical Analyses

Although the estimation method used for structural equation modeling is reported, the manuscript does not specify how missing data were handled. In addition, the authors should report whether and how the assumptions of normality were examined and whether the data met these assumptions.

7. Insufficient Evidence for Cultural Validity

The manuscript refers to the Korean floor-sitting culture as an important contextual factor; however, it is unclear how the final 12 items specifically reflect this cultural characteristic. Additional explanation or examples are needed to justify the cultural relevance of the developed scale.

8. Inconsistencies in Terminology and Language Issues

There are several inconsistencies and language-related issues throughout the manuscript, including:

• Inconsistent use of “sedentary behavior regulation”, “sedentary behavior management”, and “sedentary behaviour regulation”

• Inconsistent use of “subfactor” and “sub-factor”

• Inconsistent use of “hour” and “hours”

In addition, some sentences are verbose or grammatically awkward. Professional English language editing is strongly recommended.

9. Other Issues

• Please verify the accuracy of the data collection period and the timing of ethical approval.

• In Table 1, the CFA value for Residential area: Gyeonggi-do appears to be incorrect and should be carefully checked.

**Do you want your identity to be public for this peer review?** For information about this choice, including consent withdrawal, please see our For information about this choice, including consent withdrawal, please see our Privacy Policy .

Reviewer #1: **Yes:** Héctor González-de la TorreHéctor González-de la Torre

Reviewer #2: No

---

## [Author Response · Author response to Decision Letter 1]

17 Feb 2026

We sincerely thank the judges for their meticulous review.

We have modified the manuscript according to your suggestions, rewriting and rephrasing sections to improve clarity, adding further information, and explaining in detail the points that were previously vague. For your convenience, we have set the revisions in the manuscript in red. We believe that the revised version of this paper will interest the readership of the PLOS ONE.

We have seriously considered the reviewers’ comments and carefully revised the manuscript.

Please see the attached file for responses to the judges' comments.

---

## [Decision Letter · Decision Letter 1]

8 Mar 2026

Dear Dr. Shin,

Thank you for submitting your manuscript to PLOS ONE. After careful consideration, we feel that it has merit but does not fully meet PLOS ONE’s publication criteria as it currently stands. Therefore, we invite you to submit a revised version of the manuscript that addresses the points raised during the review process.

We look forward to receiving your revised manuscript.

Kind regards,

Muhammad Shawqi, MD, MSc

Guest Editor

PLOS One

Journal Requirements:

Additional Editor Comments:

I really appreciate the rationale behind this study aimed to develop and validate a self-report scale for assessing

sedentary behavior regulation in Korean adults. The authors did a significant research effort to address this research gap in the Korean population. It would be more beneficial for the targeted population if this fact was explicitly stated in the title of this research by adding the phrase “in Korean adults population”. Moreover, the introduction of this manuscript lacks a more detailed reflection on how this self-report scale would meet the meeds of the Korean population and help them to solve this problem. With these minor revisions along with addressing additional other minor concerns raised by the reviewers, the message will be better disseminated.

Reviewers' comments:

Reviewer's Responses to Questions

**Comments to the Author**

Reviewer #1: All comments have been addressed

Reviewer #2: (No Response)

2. Is the manuscript technically sound, and do the data support the conclusions?

Reviewer #1: Yes

Reviewer #2: Partly

3. Has the statistical analysis been performed appropriately and rigorously?

Reviewer #1: Yes

Reviewer #2: Yes

4. Have the authors made all data underlying the findings in their manuscript fully available?

Reviewer #1: Yes

Reviewer #2: Yes

5. Is the manuscript presented in an intelligible fashion and written in standard English?

Reviewer #1: Yes

Reviewer #2: Yes

Reviewer #1: Dear Authors,

Thank you for allowing me to review the revised manuscript. I believe you have made a substantial effort to implement the changes suggested by both reviewers. Overall, the manuscript has improved considerably and is now suitable for publication. I would like to highlight only two minor points that do not preclude publication:

-Line 238: “reliability was evaluated by assessing internal consistency with Cronbach’s α and test-retest reliability using a two-week interval with a subsample of 30 adults.”

Please add that McDonald’s omega was also calculated.

-Decision to remove five items: You state that “An additional five items (x13, x17, x21, x25, and x30) were removed because of cross-loadings, defined as factor loadings of .40 or higher on two factors in the pattern matrix.”

I respect this decision; however, for future studies, rather than removing such items, it may be more appropriate to re-evaluate them in relation to the theoretical framework and the measurement model. Cross-loadings are not necessarily indicative of poor item performance and may sometimes reflect phenomena such as essential unidimensionality. In future work, you may wish to consider assigning the item to the factor on which it loads most strongly, or to the dimension that best fits the item content based on the theoretical framework.

From my perspective, the manuscript can be published and does not require any further review on my part.

Reviewer #2: I appreciate the authors' detailed responses to my initial review comments. Several concerns have been addressed satisfactorily, and the manuscript has improved in many aspects. However, some issues remain insufficiently resolved, and new concerns have emerged upon closer reading of the revised text. I recommend further revision before the manuscript can be accepted.

1. Convergent Validity (AVE) (Point 1)

The addition of the HTMT analysis is a valuable supplement, and I appreciate the repositioning of the study as a "developmental validation". However, the Fornell–Larcker criterion is clearly not met for Factor 1 (√AVE = .67 < inter-factor r = .68), and only marginally met for Factor 2 (√AVE = .68 = r = .68). The current manuscript states this indicates "limited to marginal discriminant validity based on this criterion alone" (Lines 426–429), then pivots to the HTMT result as if it resolves the issue. This framing risks understating the concern. The authors should:

(a) Explicitly acknowledge that the Fornell–Larcker criterion was not satisfied, rather than using softened language such as "limited to marginal."

(b) In the Discussion/Limitations, note that the failure to meet the Fornell–Larcker criterion, combined with AVE < .50, raises the possibility that the two factors may not be sufficiently distinct at the construct level, and that this should be a priority for future refinement.

2. Theoretical Justification for the Factor Structure (Point 2)

The expanded discussion of self-awareness as a "cognitive antecedent or prerequisite" (Lines 510–526) is an improvement. However, two concerns remain:

(a) Post-hoc modification to the conceptual framework: The revised Introduction now states that "self-awareness is conceptualized not as an independent regulatory dimension but as a cognitive prerequisite that enables—rather than constitutes—the enactment of sedentary behavior regulation" (Lines 123–126). However, this reconceptualization appears to have been introduced after the factor analysis revealed that self-awareness items performed poorly. In the original conceptual framework, self-awareness was clearly positioned as one of three core components. Presenting this post-hoc reinterpretation in the Introduction as if it were part of the original theoretical framework is misleading. The authors should either: remove this reframing from the Introduction and instead present it as a theoretical insight derived from the empirical findings in the Discussion, or clearly state in the Introduction that this conceptualization was refined based on empirical evidence from the present study.

(b) Absence of "relatedness" in SDT alignment: The Discussion (Lines 491–509) explains how the two factors align with autonomy and competence, but the third basic psychological need in SDT relatedness is not addressed. Since the SBRS is explicitly grounded in SDT, the authors should discuss why relatedness does not appear to be captured by the final scale and what implications this has for the theoretical completeness of the instrument. Even a brief acknowledgment (e.g., that the individual nature of sedentary behavior regulation may explain the limited role of relatedness) would strengthen the argument.

3. Item Reduction Process (Point 3)

The manuscript now provides somewhat more detail on why items were removed during EFA (low communalities, cross-loadings) and CFA (modification indices, conceptual redundancy). However, my original request was for a list of the excluded items and their content. I understand that the authors reference Supplementary Table 2 as containing content validity and item reduction information. However:

(a) The specific content (text) of the 20 removed items and the component to which each item originally belonged (self-awareness, behavioral practice, or environmental design) should be available to reviewers and readers, either in the main text or as supplementary material. Without this information, it is impossible for readers to independently evaluate whether the removed items were appropriately excluded or whether important content was lost.

(b) In particular, since all self-awareness items were excluded, it would be informative to present these items and their psychometric performance (e.g., communalities, factor loadings) to support the claim that their removal was empirically justified.

4. Criterion Validity Interpretation (Point 5)

The revised interpretation appropriately describes the effect sizes as "small to moderate" (Lines 549–552), which is a clear improvement over the original language. However, the Abstract still states "Criterion-related validity was supported by a significant positive correlation with physical activity (r = .19, p < .001)" (Lines 33–35) without any qualifier regarding effect size. A brief modifier (e.g., "a small but significant positive correlation") in the Abstract would improve consistency with the Discussion and prevent readers from overinterpreting the association based on the Abstract alone.

Additionally, the authors argue that the small effect sizes are "theoretically meaningful, given that the SBRS is designed to assess regulatory capacity rather than direct behavioral frequency" (Lines 551–552). While this argument has some merit, it would be strengthened by citing comparable effect sizes from validation studies of other self-regulation or self-efficacy scales. Without such comparative context, the justification remains somewhat circular.

5. Cultural Validity (Point 7)

The expanded discussion (Lines 527–539) provides a general argument that the SBRS items capture floor-sitting-related regulatory behaviors indirectly. However, the argument remains speculative because:

(a) No empirical evidence is presented to demonstrate that respondents' SBRS scores differ as a function of floor-sitting frequency or that items function differently across sedentary contexts (e.g., chair-based vs. floor-based).

(b) The claim that items were "intentionally designed to capture regulatory behaviors that are prevalent in sedentary contexts characteristic of Korean daily life" (Lines 528–529) would benefit from at least one or two concrete examples mapping specific items to floor-sitting-related regulation.

I recognize that a full analysis of differential item functioning may be beyond the scope of this study, but the authors should temper their claims about cultural sensitivity by acknowledging that this remains an empirical question to be tested in future research, rather than presenting it as an established feature of the SBRS.

Minor Issues

1. Abstract, Line 37: The phrase "the SBRS proved to be a reliable and valid instrument" is strong language for a scale with sub-threshold AVE. Consider revising to "the SBRS demonstrated preliminary evidence of reliability and validity" to maintain consistency with the developmental framing adopted throughout the revised Discussion.

2. Lines 116–118: The phrase "sedentary behavior regulation is conceptualized as an autonomously motivated process involving self-awareness of sedentary patterns, active behavioral practices to interrupt or reduce sedentary time, and preventive environmental structuring" still lists three components including self-awareness, which contradicts the statement two sentences later (Lines 123–126) that self-awareness is not an independent regulatory dimension. This internal inconsistency within the Introduction should be reconciled.

3. Table 4: The standard deviation for physical activity (PA) is reported as "3901.88 ± 59.14." An SD of 59.14 for a variable with a range of 0–46,350 MET-min/week appears very small. Please verify whether this is a typographical error.

4. Line 609: The Conclusions state "developed and preliminarily validated," which is appropriate and consistent with the revised framing. However, Line 617 states "providing a valid measurement tool," which reverts to a more definitive claim. This should be softened to maintain consistency.

**Do you want your identity to be public for this peer review?** For information about this choice, including consent withdrawal, please see our For information about this choice, including consent withdrawal, please see our Privacy Policy .

Reviewer #1: **Yes:** Héctor González-de la TorreHéctor González-de la Torre

Reviewer #2: No

---

## [Author Response · Author response to Decision Letter 2]

25 Mar 2026

We sincerely appreciate the Editor and the reviewers for their careful reading of our manuscript and for their thoughtful and constructive comments. Their detailed suggestions have substantially strengthened the quality and clarity of this work. We are truly grateful for the time and scholarly effort they invested in helping us improve the manuscript.

---

## [Editor Report · Decision Letter 2]

26 Mar 2026

Development and Validation of the Sedentary Behavior Regulation Scale in Korean Adults Population

PONE-D-25-60930R2

Dear Dr. Shin,

We’re pleased to inform you that your manuscript has been judged scientifically suitable for publication and will be formally accepted for publication once it meets all outstanding technical requirements.

Kind regards,

Muhammad Shawqi, MD, MSc

Guest Editor

PLOS One

Additional Editor Comments:

The authors did a significant improvement to the manuscript by positively taking the raised comments into consideration. I appreciate the authors’ updates and clarifications, and hope that this scale will be used among the targeted population to encourage further research and development.

---

## [Editor Report · Acceptance letter]

PONE-D-25-60930R2

PLOS One

Dear Dr. Shin,

I'm pleased to inform you that your manuscript has been deemed suitable for publication in PLOS One. Congratulations! Your manuscript is now being handed over to our production team.

Kind regards,

on behalf of

Dr. Muhammad Shawqi

Guest Editor

PLOS One